# Role of Klotho and AGE/RAGE-Wnt/β-Catenin Signalling Pathway on the Development of Cardiac and Renal Fibrosis in Diabetes

**DOI:** 10.3390/ijms24065241

**Published:** 2023-03-09

**Authors:** Beatriz Martín-Carro, Julia Martín-Vírgala, Sara Fernández-Villabrille, Alejandra Fernández-Fernández, Marcos Pérez-Basterrechea, Juan F. Navarro-González, Javier Donate-Correa, Carmen Mora-Fernández, Adriana S. Dusso, Natalia Carrillo-López, Sara Panizo, Manuel Naves-Díaz, José L. Fernández-Martín, Jorge B. Cannata-Andía, Cristina Alonso-Montes

**Affiliations:** 1Bone and Mineral Research Unit, Instituto de Investigación Sanitaria del Principado de Asturias (ISPA), Hospital Universitario Central de Asturias, 33011 Oviedo, Spain; 2Redes de Investigación Cooperativa Orientadas a Resultados en Salud (RICORS), RICORS2040 (Kidney Disease), Instituto de Salud Carlos III, 28029 Madrid, Spain; 3Laboratory of Medicine, Hospital Universitario Central de Asturias, 33011 Oviedo, Spain; 4Unit of Cell Therapy and Regenerative Medicine, Hematology and Hemotherapy Service, Instituto de Investigación Sanitaria del Principado de Asturias (ISPA), Hospital Universitario Central de Asturias, 33011 Oviedo, Spain; 5Research Unit, Hospital Universitario Nuestra Señora de Candelaria, 38010 Santa Cruz de Tenerife, Spain; 6Nephrology Service, Hospital Universitario Nuestra Señora de Candelaria, 38010 Santa Cruz de Tenerife, Spain; 7Division of Endocrinology, Metabolism and Lipid Research, Washington University School of Medicine, St. Louis, MO 63110, USA; 8Department of Medicine, Universidad de Oviedo, 33006 Oviedo, Spain; 9Department of Psychobiology, Universidad de Oviedo, 33003 Oviedo, Spain

**Keywords:** sKlotho, sRAGE, AGEs, diabetes, T1DM, fibrosis, Wnt/β-catenin, kidney, heart

## Abstract

Fibrosis plays an important role in the pathogenesis of long-term diabetic complications and contributes to the development of cardiac and renal dysfunction. The aim of this experimental study, performed in a long-term rat model, which resembles type 1 diabetes mellitus, was to investigate the role of soluble Klotho (sKlotho), advanced glycation end products (AGEs)/receptor for AGEs (RAGE), fibrotic Wnt/β-catenin pathway, and pro-fibrotic pathways in kidney and heart. Diabetes was induced by streptozotocin. Glycaemia was maintained by insulin administration for 24 weeks. Serum and urine sKlotho, AGEs, soluble RAGE (sRAGE) and biochemical markers were studied. The levels of Klotho, RAGEs, ADAM10, markers of fibrosis (collagen deposition, fibronectin, TGF-β1, and Wnt/β-catenin pathway), hypertrophy of the kidney and/or heart were analysed. At the end of study, diabetic rats showed higher levels of urinary sKlotho, AGEs and sRAGE and lower serum sKlotho compared with controls without differences in the renal Klotho expression. A significant positive correlation was found between urinary sKlotho and AGEs and urinary albumin/creatinine ratio (uACR). Fibrosis and RAGE levels were significantly higher in the heart without differences in the kidney of diabetic rats compared to controls. The results also suggest the increase in sKlotho and sRAGE excretion may be due to polyuria in the diabetic rats.

## 1. Introduction

The global prevalence of diabetes has tripled in the last 20 years, and it is projected to increase by ∼50% by the year 2045 [1]. Diabetes-related complications, especially those related to the cardiovascular and renal system, compromise the quality of life of these patients and increase the premature mortality rate [2,3].

Fibrosis plays an important role in the pathogenesis of diabetic complications and contributes to the development of cardiac and renal dysfunction [4,5]. However, diabetes-associated fibrosis has not been fully elucidated regardless of its clinical relevance.

In diabetes, the underlying signalling pathways involved in fibrosis are highly complex, with a wide range of functional drivers. Among them, increased advanced glycation end products (AGEs) and their receptor RAGE play an important role in renal and cardiac fibrosis by the activation of several overlapping fibrotic pathways [6,7].

Among the different mediators involved in the diabetic cardiomyopathy and nephropathy, several experimental studies have linked the Wnt/β-catenin signalling pathway with the development and progression of cardiac and renal fibrosis [8,9]. The Wnt/β-catenin is an evolutionarily conserved signalling pathway involved in heart and kidney development; however, it is functionally silent in adult tissues in which its activation is linked to the ageing- and injury-related fibrosis [10]. In addition, in diabetes the activity of the Wnt pathway could be modulated by the Klotho levels.

Klotho is an anti-ageing protein predominantly expressed in renal tubular epithelial cells [11], which can be presented as transmembrane protein and a soluble form (sKlotho), that can be found in blood and urine, which plays a critical role in phosphorus homeostasis, acting as a humoral factor with pleiotropic effects on multiple organs [12,13]. Previous experimental studies have shown that Klotho exerts a protective effect on both the heart and kidneys [14]. This protection has been related with the inhibition of the Wnt pathway through protein–protein interactions between sKlotho and the Wnt pathway activators [15,16]. Furthermore, Klotho has been postulated as a biomarker of progression of heart and kidney damage [17,18,19]. The studies of Klotho in diabetes, mainly in type 1 diabetes mellitus (T1DM), are scarce, showing controversial results [20,21,22,23]. In addition, the relationship between Klotho and the AGE/RAGE pathway has not been studied in animal models of diabetes.

Thus, the aim of this study was to investigate: (a) The Klotho levels (soluble and transmembrane protein in kidney) and its relationship to fibrosis; and (b) If Wnt/β-catenin and AGE/RAGE pathways are involved in heart and kidney fibrosis in a long-term (24 weeks) rat model that resembles the T1DM.

## 2. Results

### 2.1. General, Glycaemic, Renal, and Cardiac Biochemical Parameters

Twenty-four weeks after receiving streptozotocin (STZ), the rats showed significant lower body weight (BW) and higher urine volume compared to controls (Table 1). Diabetic rats showed higher levels of serum and urinary glucose, glycosylated haemoglobin (Hb1Ac), plasma and urinary AGEs and urinary soluble RAGE (sRAGE), but significantly lower levels of plasma sRAGE (Table 1). The urinary AGEs were significantly higher than in Week 9 (Appendix A). Linear regression analysis showed a positive correlation between urinary AGEs and serum glucose on Week 24 (r = 0.839; *p* < 0.001), which persisted after adjustment for diabetes (r = 0.859; *p* < 0.001) (Appendix A).

The kidney analysis showed that the ratio kidney BW (left and right) was higher in the diabetic rats (Table 1). Rats that received STZ showed significantly higher levels of serum creatinine and urea, as well as urinary albumin/creatinine ratio (uACR) without changes in creatinine clearance, while serum levels of protein and albumin were significantly lower (Table 1). The uACR was significantly higher from Week 9 to Week 24 (Appendix A). Compared with control, diabetic rats also showed a seven-fold higher level of urinary calcium without changes in serum calcium (Table 1). The urinary calcium was significantly higher on Week 9 (Appendix A).

The heart study showed a significative increase in the heart BW ratio and in the plasma levels of N-terminal pro-B-type natriuretic peptide (NT-proBNP) in diabetic rats compared to controls (Table 1).

Serum sKlotho was significantly lower and urinary sKlotho significantly higher in diabetic rats (Table 1), the differences in urinary sKlotho increased at the end of the study (Figure 1A). A significant positive correlation was found between urinary sKlotho and urinary AGEs and uACR on Week 24 after adjusting for diabetes (Figure 1B,C). No differences were found in the renal Klotho expression, neither in mRNA nor in protein levels (Figure 2).

### 2.2. Diabetes-Related Kidney Changes towards Fibrosis

#### 2.2.1. Histological Analysis

The kidneys showed that the median values of the diameters of the proximal tubules were higher in diabetic rats (47.19 [45.87–49.50] vs. 54.89 [53.02–56.36] µm; *p* < 0.001) (Figure 3A). No changes in collagen deposition stained by Picrosirius Red were detected (Figure 3B). The electron microscopy examination showed a filtration barrier with thickened glomerular basement membrane and denuded foot processes of the podocytes in the glomeruli of the kidneys from diabetic rats, and pyknotic nuclei were also found (Appendix A).

#### 2.2.2. Molecular Markers of Fibrosis, Wnt/β-Catenin, RAGE and ADAM10

Despite having no differences in renal collagen deposition between the two groups, higher mRNA levels of fibronectin (0.89 [0.62–1.39] vs. 1.32 [1.15–1.45] R.U.; *p* < 0.05) (Appendix A) and no changes in *Tgf-β1* were observed in diabetic rats compared to controls.

In the kidney of diabetic rats, the *Dkk1* gene (0.85 [0.68–1.11] vs. 1.80 [1.48–3.62] R.U.; *p* < 0.001) and protein (100.17 [82.24–104.58] vs. 114.02 [104.62–136.81] %; *p* < 0.05) expression were higher compared with controls (Appendix A), without differences in the gene expression of *Sost* and *Sfrp4*; meanwhile *Sfrp2* was significantly lower (0.72 [0.58–1.19] vs. 0.30 [0.20–0.61] R.U.; *p* < 0.01) (Appendix A). No differences were found in the protein levels of active β-catenin.

Although there were no differences in the *Rage* gene expression between the two groups, the protein levels (100.73 [73.72–116.78] vs. 119.62 [110.71–145.93] %; *p* < 0.05) were increased in the diabetic group. Additionally, the gene expression of *Adam10* (0.95 [0.76–1.18] vs. 1.34 [0.94–1.45] R.U.; *p* < 0.05) was increased in the kidney of diabetic rats compared with controls (Appendix A).

### 2.3. Diabetes-Related Cardiac Changes towards Fibrosis

#### 2.3.1. Histological Analysis

The cardiomyocyte size was significantly higher (22.06 [20.66–23.38] vs. 25.29 [24.51–25.95] µm; *p* < 0.001) (Figure 4A) and deposition of collagen increased (1.96-fold), (2.61 [1.75–3.23] vs. 5.19 [4.49–6.49]%; *p* < 0.001) in diabetic rats (Figure 4B).

#### 2.3.2. Molecular Markers of Fibrosis, Wnt/β-Catenin RAGE and ADAM10

The gene expression of fibrosis markers, such as fibronectin (0.99 [0.90–1.07] vs. 1.80 [1.32–2.27] R.U.; *p* < 0.01) and *Tgf-β1* (0.99 [0.87–1.16] vs. 1.18 [1.09–1.30]), were significantly higher in diabetic rats (Appendix A).

High glucose levels maintained for 24 weeks resulted in a higher gene expression of Wnt pathway inhibitors *Dkk1* (0.81 [0.56–1.01] vs. 2.11 [1.46–3.24] R.U.; *p* < 0.05) and *Sfrp2* (0.51 [0.33–1.30] vs. 1.75 [0.85–2.51] R.U.; *p* < 0.01) compared to controls (Appendix A), without changes in *Sost* and *Sfrp4* expression. The diabetic rats also showed significantly higher protein levels of DKK1 (95.04 [84.57–108.49] vs. 119.17 [105.20–145.45] %; *p* < 0.05) and active β-catenin (95.34 [81.33–114.87] vs. 126.04 [103.51–177.54] %; *p* < 0.05) (Appendix A).

The mRNA (0.8776 [0.7504–1.2155 vs. 1.353 [1.227–1.575] R.U.; *p* < 0.001) and protein (99.59 [95.58–104.45] vs. 110.1 [105.9–119.6] %; *p* < 0.01) levels of the *Rage* and *Adam10* (1.048 [0.87–1.09] vs. 1.72 [1.50–2.13] R.U.; *p* < 0.001) gene expression levels were significantly higher in diabetic rats compared with controls (Appendix A–I).

## 3. Discussion

The present study used a novel long-term (24 weeks) rat model of STZ-induced diabetes with insulin administration, which successfully resembles chronic T1DM characterized by weight loss, polyuria, hyperglycaemia, hyperglycosuria, and higher HbA1c. At the end of study, diabetic rats showed higher levels of urinary sKlotho, urinary AGEs and uACR, while serum sKlotho was lower compared with controls. There were no differences in the renal expression of Klotho, suggesting that the lower serum sKlotho found in the diabetic rats could be due to urinary loss as consequence of polyuria (19-fold higher urinary volume), rather than to a lower kidney expression. The lower serum levels of sKlotho due to its higher excretion could be related to the increase AGE/RAGE levels and the activation of pro-fibrotic Wnt/β-catenin pathway, observed mainly in the heart.

Diabetes is associated with a greater risk of cardiovascular complications and kidney damage that leads to chronic heart and kidney failure, resulting in a bidirectional disorder known as cardiorenal syndrome [2,3]. Fibrosis, besides being a marker of injury progression, has been proposed as a driver of the pathophysiology of the cardiorenal syndrome, but the timing of its progression throughout the course of diabetes, as well as its association with other mediators, such as sKlotho and other important molecules measured in this study, are critically important for future preventive strategies.

Few studies have analysed the changes of both, serum and urinary sKlotho in T1DM. Opposite to the results of the present work, a previous study using a similar rat model but only during 14 days of diabetes, found higher serum sKlotho and lower urine sKlotho and renal Klotho expression [24]. The difference between both studies could be due to the time and degree of renal damage induced by the diabetes. The mentioned study showed that albuminuria was much higher than in the present study (2.57 ± 0.67 mg/24 h vs. 0.67 ± 0.27 mg/24 h), with similar polyuria, suggesting a greater short-term renal damage. Meanwhile, in our long-term study, the renal damage could be considered “mild-moderate” and classified as Class I diabetic nephropathy, according to the Renal Pathology Society [25]. The rat model used in the present study produced a very mild renal function decline, and it could mimic what occurs in humans in the early phases of diabetes. In fact, and according to our results, a decrease in serum sKlotho has been reported in diabetic patients in the early stages of chronic kidney disease (CKD), while increased levels are observed thereafter [22]. The decline in sKlotho in the advanced stages of CKD has been related to the downregulation of Klotho kidney production [24,26]. However, this is not the case in our study, in which the diabetic rats showed almost no decrease in the glomerular filtration rate (GFR), as it occurred in the early stages of CKD, where the kidney involvement is only suspected by the appearance of albuminuria but not due to relevant reductions of the GFR.

No association was found between serum and urinary sKlotho levels, suggesting that urinary sKlotho is the result of a complex process [27]. Physiologically, circulating sKlotho passes from the basal to the apical side of the proximal tubular cells through transcytosis process and is eliminated in the urine, but it can also be secreted from tubules cells by sheddases [28]. As it has been previously described in an experimental model of acute kidney damage [29], tubular epithelial cells can lose their brush border membrane leading to an increase in Klotho shedding to the lumen. In the present study, it cannot be established whether hyperglycaemia and the consequent hyperfiltration enhanced Klotho transcytosis; however, the tubular histological alterations and the higher renal gene expression of *Adam10* suggest that the elevated urinary sKlotho could be a marker of the diabetes-induced inflammation and tubular damage.

Cross-sectional studies in diabetic patients, mainly in type 2 diabetes mellitus (T2DM), have shown that serum sKlotho may play a relevant role in albumin homeostasis; however, only a few of them have found a correlation between serum sKlotho and uACR [20]; in fact, little is known about the possible relationship between urinary sKlotho and the excretion of albumin. In the present study, the diabetic status induced increase in both, urinary sKlotho levels (Figure 1A) and uACR (Appendix A), throughout the whole duration of the study. This relationship was consistent, furthermore, at the end of the study, a positive significant correlation between these two parameters that persisted after adjustment for diabetes was found (Figure 1C). These results suggest that in the early phases of the diabetic kidney disease, with preserved GFR, urine sKlotho may be an earlier marker of kidney damage than serum sKlotho. However, the role of sKlotho in CKD is very complex and its possible use as an urinary marker of renal function has been poorly studied [23].

Previous studies in T1 and T2DM found a negative correlation between serum sKlotho levels and HbA1c level, one of the most studied glycated proteins and a good marker of diabetes [20,30], suggesting a close relationship between sKlotho and hyperglycaemia. In line with these findings, to the best of our knowledge, our results showed for the first time, a significant positive correlation between urinary sKlotho and urinary fluorescent AGEs, providing additional evidence for the relationship between sKlotho and diabetes. The strength of the correlation between urinary sKlotho and fluorescent AGEs (Figure 1B) suggest that the latter, an easier and quicker method to estimate sKlotho, could be used at least in the early stages of diabetes.

The decrease in sKlotho levels and the increase in circulating AGEs due to the hyperglycaemia could play a role in the diabetic complications. It is known that AGEs formation includes many heterogeneous chemical structures which are increased in diabetes and can play a relevant pathophysiological role acting directly or via a receptor-mediated (RAGE) signalling [31]. In addition, it has been shown that serum sRAGE was lower in diabetic rats, despite a higher gene expression of *Adam10*, also involved in the cleavage of membrane-bound RAGE [32]. Previous studies found increased levels of serum sRAGE in diabetic patients [33,34]; in contrast, other studies found a decreased levels of circulating sRAGE in patients with T1 and T2DM [35,36]. In some cases, this discrepancy could be explained as a consequence of differences in renal function [34]. Analyses of serum and urinary sRAGE levels in the early stages of diabetic nephropathy are scarce. In the present study, serum levels of sRAGE were lower in diabetic rats that could be explained by the increase in sRAGE excretion, similar to that observed with sKlotho. The function of sRAGE is also controversial, it has been postulated that increased sRAGE production could be related to sustained inflammation [37] but there is a broad agreement that it plays a protective role acting as a decoy receptor for AGEs [32,35]. Thus, in diabetic rats, the decrease in sRAGE and the increase in serum AGEs could increase the AGE binding to RAGE, stimulating several fibrotic pathways [6,7]. In the present study, AGE/RAGE pathway activation occurred mainly in the heart, where the expression of RAGE was significantly higher measured at both mRNA and protein levels.

In the kidney of diabetic rats, structural changes in tubules and glomerulus were associated with hyperfiltration without changes in fibrosis. By contrast, the hearts of the diabetic rats showed hypertrophy of the cardiomyocytes and increased fibrosis. These observed morphological changes are supported by the high expression of *Tgf-β1*, fibronectin and the active involvement of the Wnt/β-catenin pathway, which can trigger inhibitory and compensatory signals such as, *Dkk1* and *Sfrp2* [38,39]. In fact, although *Sfrp2* has traditionally been considered a Wnt inhibitor, its activation has been associated with the increase in the activity of β-catenin pathways, which in the heart of diabetic rats, could have triggered the activation of β-catenin, leading to increased myocardial fibrosis [39]. The structural alterations observed in the heart of the diabetic rats could be also associated with the increase in NT-pro-BNP, a biomarker of the myocardial function predictor of cardiovascular events in patients with T2DM [40].

One limitation of the study is that although the diabetic rat models used could resemble important aspects of the human diabetic disease type 1, the fibrotic changes observed in the rodent models of diabetes are mild compared to the extensive and widespread fibrosis found in patients with long-standing diabetes.

In summary, the difference in the fibrosis observed in the kidney and heart of the diabetic rats could be related to the Klotho and AGE/RAGE pathway levels. In the kidney—the main source of Klotho—Klotho expression was maintained at the same level as in the control rats and it could have played its known antifibrotic action, repressing the activation and translocation of β-catenin to the nucleus [41], though the increase in the RAGE expression levels was not clear. In the heart, the decrease in serum sKlotho—known to have a negative impact on cardiac function [42,43]—and the increase in RAGE expression, possibly related to the activation of the Wnt/β-catenin signalling pathway, may have promote the fibrotic process and the progression of the cardiac alterations observed in the diabetic rats.

## 4. Materials and Methods

### 4.1. Experimental Model

Four month-old male Wistar rats weighing 425 ± 43 g were kept under conventional conditions in the Animal Facility of the University of Oviedo with free access to water and standard food. Experimental procedures were approved by the Ethics Committee for laboratory animals of the Oviedo University.

Type 1 Diabetic Model

Thirty-four animals, housed three per cage, were randomly divided into two groups receiving either a single intraperitoneal injection of 55 mg/kg BW of freshly prepared STZ (Sigma-Aldrich, St. Louis, MO, USA) in 0.1M citrate buffer pH = 4.5 (Diabetic rats; n = 17), or citrate buffer alone (Control rats; n = 17) after 6 h of fasting under light anaesthesia with isoflurane 2%. Drinking water was supplemented with 10% sucrose solution for 24 h to prevent hypoglycaemic shock.

Tail vein blood glucose and ketone bodies were tested after 24 h of STZ administration, and daily during the six days, using a Freestyle Optimum Neo device (Abbot Diabetes Care, Witney, UK). Subcutaneous long-acting biosynthetic human insulin (Lantus^®^, Aventis Pharma, Bad Soden, Germany) (1–2 IU) was administered when blood glucose > 500 mg/dL and/or ketone bodies > 3 mmol/L.

After one week, the rats who had a blood glucose of >350 mg/dL for three consecutive days were considered “diabetic rats” and included in the study. Glycaemia and BW were monitored twice weekly. Those rats with blood glucose higher than 500 mg/dL (the upper limit of quantification of the glucometer) and significant body weight loss (>10%) received subcutaneous injections of 1–2 IU long-acting biosynthetic human insulin or subcutaneous insulin pellets (0.5 IU/24 h slow-release) (Linshin, Toronto, ON, Canada).

From Week 9 to 24, the rats were placed in metabolic cages for 24-h urine collection every 3 weeks and the day before the sacrifice. At the end of the study, the rats were anesthetized with isoflurane and sacrificed by exsanguination. Sections of the hearts and kidneys were weighed and fixed in 4% formaldehyde for histological analyses or stored at −80 °C for mRNA and protein extraction.

### 4.2. Biochemical Analyses

The blood was collected in tubes with or without EDTA and centrifuged at 3000 rpm for 15 min at 4 °C to obtain plasma or serum, respectively, and stored at −80 °C until analysis. The urine volume was measured and centrifuged for 5 min at 2500 rpm and the clear supernatant stored at −80 °C. Serum and urinary glucose, calcium, creatinine, total proteins, albumin, and serum urea were measured using a multi-channel auto-analyser (Hitachi 717; Boehringer Mannheim, Mannheim, Germany).

Commercial enzyme-linked immunosorbent assay (ELISA) kits were performed to analysed the levels of AGEs (CSB-E09413r, Cusabio, Houston, TX, USA), sRAGE (MBS029347, MyBioSource, San Diego, TX, USA), and NT-proBNP (CSB-E08752r, Cusabio, Houston, TX, USA) in serum/plasma and/or urine, following manufacturer’s protocols. Fluorescent urine AGEs were also measured by a fluorometric measurement method, following a previously described procedure [44].

Serum sKlotho was measured by an immunoprecipitation–immunoblot assay at the O’Brien Kidney Research Centre in UT Southwestern, and urinary sKlotho was determined at our laboratory following the protocol of the O’Brien Kidney Research Center [45], briefly summarized below in the immunoblotting section.

Urinary biomarkers were showed as a ratio of the urinary creatinine. Creatinine clearance and uACR were calculated using the following formulas: creatinine clearance (mL/min) = urinary creatinine (mg/dL) × urinary volume (mL)/serum creatinine (mg/dL) × 1440 (min); uACR (mg/g) = urinary albumin (mg/L)/urinary creatinine (g/L). Blood Hb1Ac was measured at sacrifice using A1Cnow+^®^ (PTS Diagnostics, Indianapolis, IN, USA).

### 4.3. Histological Analyses

The hearts and kidneys were formalin-fixed and paraffin-embedded. Tissue sections of 5 µm were stained with Picrosirius Red and periodic acid–Schiff following standard protocols to assess the degree of fibrosis, as well as cardiomyocyte and proximal tubules sizes. For this purpose, 5 and 10 random images were acquired under a light microscope (DMRXA2, Leica Microsystems, Wetzlar, Germany) equipped with a Leica DFC7000 T camera (Leica Microsystems). ImageJ software was used for treatment and analyses of acquired images.

Kidney fibrosis was assessed by measuring the percentage of tissue area stained by Picrosirius Red divided by total tissue area at a magnification of 10× excluding perivascular fibrosis and kidney glomeruli. The length of the short axis of proximal tubules in the glomerulus was also measured at a magnification of 20×. Cardiomyocyte hypertrophy was assessed by measuring the cross-sectional widths of cardiomyocytes drawing a line that crossed the center of the nucleus at a magnification of 40×. A minimum of 150 proximal tubules and cardiomyocytes were measured by tissue section.

For electron microscopy analysis, 7 kidney samples (3 controls and 4 diabetic rats) were randomly selected, the pieces (3 mm diameter) were immediately fixed in glutaraldehyde at room temperature for 3 h. After processing and embedding, the semithin (1 μM) sections were stained with toluidine blue for glomerular location. Ultrathin (200 Å) sections were collected, stained with uranyl acetate and lead citrate, and examined in a JEOL 1011 transmission electron microscope.

### 4.4. RNA Extraction and Quantitative Real-Time PCR

Total RNA was extracted from the kidneys and hearts using TRI-Reagent (Sigma-Aldrich, St. Louis, MO, USA). cDNA was then synthesized from 1 µg of total RNA using a high-capacity cDNA reverse transcription kit (Applied Biosystems, Waltham, MA, USA). To measure Dickkopf-related protein 1 (*Dkk1*) (Rn01501537, Thermo Fisher Scientific, Waltham, MA, USA), sclerostin (*Sost*) (Rn00577971), secreted frizzled-related protein 2 (*Sfrp2*) (Rn01458837), *Sfrp4* (Rn00585549), fibronectin (Rn00569575), transforming growth factor beta 1 (*TGF*-*β1*) (Rn00572010), receptor for advanced glycation end products (*Rage*) (Rn00584249), a disintegrin and metalloproteinase domain-containing protein 10 (*Adam10*) (Rn01530753), and Klotho (Rn00580123) mRNA levels quantitative real-time PCR was used.

TaqMan Universal PCR Master Mix (Thermo Fisher Scientific, Waltham, MA, USA) was used for the amplification of target genes according to the manufacturer’s protocol in a QuantStudio 3 Real-Time PCR System (Applied Biosystems, Waltham, MA, USA). Glyceraldehyde-3-phosphate dehydrogenase (*Gapdh*) (Rn99999916) was used for normalization. The ∆∆CT method was used to quantify the relative expression of each gene [46].

### 4.5. Immunoblotting

Kidney and heart tissues were homogenized in a RIPA buffer, and the total protein content was measured by the DC protein assay reagents (Bio-Rad, Hercules, CA, USA). Proteins were separated by sodium dodecyl sulphate–polyacrylamide gel (8%) electrophoresis (SDS-PAGE). Urine samples were loaded into commercial NuPAGE 4–12% Bis-Tris gels (Thermo Fisher Scientific, Waltham, MA, USA). In both cases, the proteins were transferred onto polyvinylidene difluoride (PVDF) membranes (Amersham Hybond, Amersham Biosciences, Amersham, UK). Blotting efficiency was checked by Ponceau red dyeing (Sigma-Aldrich, St. Louis, MO, USA). The membranes were incubated with specific antibodies following the manufacturer’s instructions: Non-phospho (Active) β-Catenin (#8814, dilution 1:1000; Cell Signaling, Danvers, MA, USA), DKK1 (MAB1765, dilution 1:500; R&D Systems, Minneapolis, MN, USA), RAGE (PA5-78736, 0.5 µg/mL; Thermo Fisher Scientific, Waltham, MA, USA) and Klotho (#KO603, dilution 1:1250; Trans Genic Inc., Chuo-ku, Japan).

Anti-rabbit, anti-rat, anti-goat, or anti-mouse (Santa Cruz, Dallas, TX, USA) were used as secondary antibodies detected with the ECL Western Blotting Detection Kit (Bio-Rad) and the ChemiDoc Gel Imaging System Model XRS (Bio-Rad), and were quantified using Quantity One 1-D Analysis Software Version 4 (Bio-Rad). All blots were rehybridized with glyceraldehyde-3-phosphate dehydrogenase (GAPDH; dilution 1:3000; Santa Cruz Biotechnology) for normalization.

### 4.6. Statistical Analysis

Results were expressed as median and interquartile ranges. The differences between groups were assessed using the non-parametric Wilcoxon rank sum test. Linear regression was used to assess the correlation between continuous variables. Statistically significant differences were considered when *p* < 0.05. The statistical analysis was carried out using R software for windows (Version 4.1.2).

## Figures and Tables

**Figure 1 ijms-24-05241-f001:**
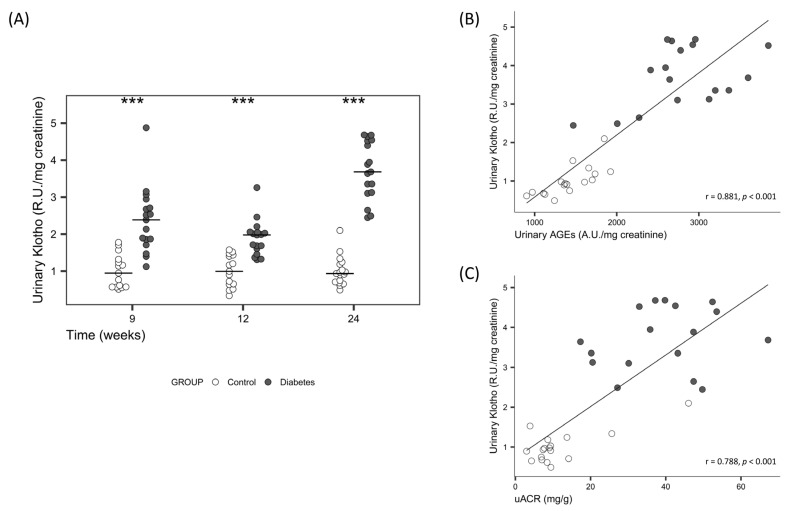
(**A**) Time course of urine sKlotho protein levels on Week 9, 12 and 24 in controls and diabetic rats. Non-parametric Wilcoxon rank sum test was used. Horizontal line represents median. *** *p* < 0.001 vs. control. Correlation between urinary sKlotho and (**B**) urinary AGEs and (**C**) uACR, in Week 24. Linear regression was used. R.U.: relative units; uACR: urinary albumin/creatinine Ratio; AGEs: advanced glycation end products.

**Figure 2 ijms-24-05241-f002:**
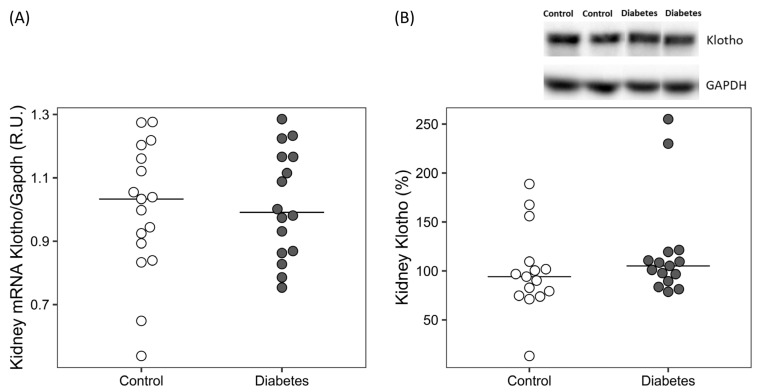
Kidney Klotho levels in the diabetic rats. (**A**) Gene expression and (**B**) relative quantification and representative image of Western blot analysis. R.U.: relative units. Non-parametric Wilcoxon rank sum test was used. Horizontal line represents median.

**Figure 3 ijms-24-05241-f003:**
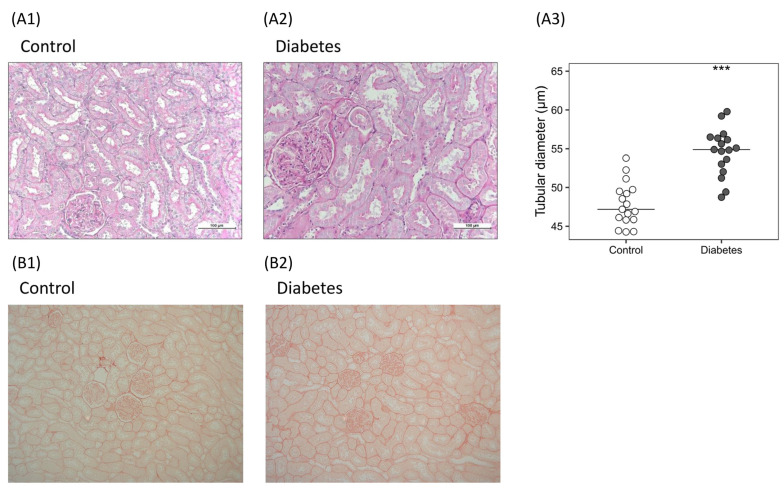
Histological findings in the kidneys of the diabetic rats. (**A**) Renal hypertrophy: representative images of histological sections stained with periodic acid–Schiff in (**A1**) control and (**A2**) diabetic rats, and quantification of (**A3**) proximal tubules diameter. Non-parametric Wilcoxon rank sum test was used. Horizontal line represents median. *** *p* < 0.001 vs. control. (**B**) Kidney fibrosis: representative images of histological sections stained with Picrosirius Red in (**B1**) control and (**B2**) diabetic rats.

**Figure 4 ijms-24-05241-f004:**
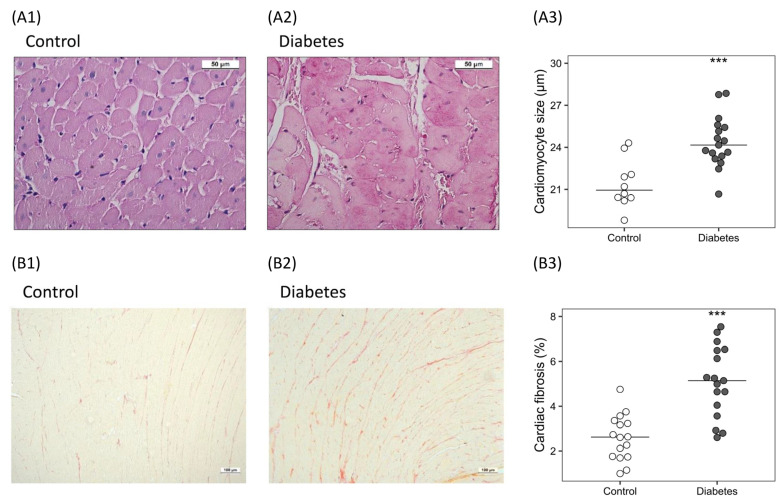
Histological changes in the hearts of the diabetic rats. (**A**) Cardiac hypertrophy: representative images of histological cross sections stained with periodic acid–Schiff in (**A1**) control and (**A2**) diabetic rats, and (**A3**) relative quantification of cardiomyocyte size. (**B**) Cardiac fibrosis: representative images of histological sections stained with Picrosirius Red in (**B1**) control and (**B2**) diabetic rats, and (**B3**) percentage of cardiac fibrosis. Non-parametric Wilcoxon rank sum test was used. Horizontal line represents median. *** *p* < 0.001 vs. control.

**Table 1 ijms-24-05241-t001:** Relevant general, biochemical, renal and cardiac parameters from control and diabetic rats at the end of the study.

	Control (n = 17)	Diabetes (n = 17)	*p*-Value
Body weight (g)	541 [474, 580]	384 [362, 405]	<0.001
Urine volume (mL)	8 [7, 13.5]	150 [129, 172]	<0.001
**Glycaemic parameters**			
Serum Glucose (mg/dL)	193 [182.2, 204.8]	714.4 [682.8, 731.8]	<0.001
HbA1c (%)	5.0 [4.7, 5.0]	>13 *	<0.001
Plasma AGEs (µg/mL)	155.8 [99.9, 530.0]	983.3 [761.6, 1115.5]	<0.001
Plasma sRAGE (ng/mL)	8.3 [7.2, 9.7]	5.2 [4.9, 7.0]	0.003
Urinary Glucose (mg/24 h)	4.5 [3.42, 6.46]	15,596.7 [13,680.0, 17077.5]	<0.001
Urinary AGEs (R.U./mg creatinine)	1393.2 [1241.4, 1659.2]	2739.7 [2594.3, 3125.0]	<0.001
Urinary sRAGE (ng/mg creatinine)	2.9 [2.2, 3.7]	43.88 [36.6, 50.0]	<0.001
**Renal parameters**			
Kidney weight (mg/100 g BW)			
Right	248 [237, 257]	427 [391, 451]	<0.001
Left	235 [218, 262]	406 [376, 440]	<0.001
Serum Proteins (g/L)	61.5 [59.0, 64.8]	55.5 [54.4, 57.6]	<0.001
Serum Albumin (g/L)	42 [39.8, 44.1]	35 [33.6, 36]	<0.001
Serum Creatinine (mg/dL)	0.3 [0.2, 0.3]	0.3 [0.3, 0.4]	0.009
Serum Urea (mg/dL)	25.9 [23.4, 27.6]	34.3 [31.3, 36.1]	<0.001
Serum Calcium (mg/dL)	10.1 [9.9, 10.2]	10.1 [9.7, 10.3]	0.654
Serum sKlotho (pM)	364.9 [328.9, 444.2]	321.3 [292.7, 352.3]	0.024
Urinary proteins (mg/mg creatinine)	0.7 [0.7, 0.8]	1.0 [1.0, 1.2]	<0.001
uACR (mg/g)	8.6 [7.1, 9.4]	39.7 [30.1, 47.4]	<0.001
Creatinine clearance (mL/min)	3.1 [2.6, 3.4]	3.0 [2.7, 3.8]	0.480
Urinary calcium (mg/mg creatine)	0.2 [0.22–0.28]	1.9 [1.6–2.37]	<0.001
Urinary sKlotho (R.U./mg creatinine)	0.9 [0.7, 1.1]	3.6 [3.1, 4.5]	<0.001
**Cardiac parameters**			
Heart weight (mg/100 g BW)	226 [210, 233]	242 [230, 255]	0.006
NT-proBNP (µg/mL)	269.7 [209.6, 296.1]	302.1 [286, 334.1]	0.017

* Above the quantification limit. Values are expressed as median [IQR]; Hb1Ac: Glycosylated haemoglobin; AGEs: advanced glycation end products; sRAGE: soluble receptor for AGEs; R.U.: relative units; BW: body weight; uACR: urinary albumin/creatinine ratio; NT-proBNP: N-terminal pro-B-type natriuretic peptide.

## Data Availability

The data presented in this study are available on request from the corresponding authors.

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
