# Peer review of "Role of Klotho and AGE/RAGE-Wnt/β-Catenin Signalling Pathway on the Development of Cardiac and Renal Fibrosis in Diabetes"

_ijms, 2023, doi:10.3390/ijms24065241_

Round 1

Reviewer 1 Report

There is a very complex and well design experimental work. In the reviewer’s opinion, because a lot of parameters were measured and evaluated, some topics are “superficial” discussed. If the experiment was conducted about 24 weeks, it would be interesting and important to evaluate modification in time of the parameters (not only for creatinine and urinary sKlotho), even it would have been less.

Row 343 – space between number and unit temperature

Row 347 – formatting “4oC”

The glycaemia was kept between 300-500 mg/dL range, in the Table 1 show a mean value of 714 mg/dL.

In the material and method section id not specified how the blood sample collection were made at weeks 9 and 12.

Rows 280 and 283: the sentences are not finished

Discussion:

-          according to other data (DOI: 10.1007/s11010-018-3400-2), the plasma level of sRAGE is elevated in diabetes, contrary to the results presented in this paper, possible explanations are needed.

-          the role and urinary level of Klotho is very complex, more detailed information are needed (doi.org/10.3389/fphar.2022.931746)

Author Response

Reviewer 1

There is a very complex and well design experimental work. In the reviewer’s opinion, because a lot of parameters were measured and evaluated, some topics are “superficial” discussed. If the experiment was conducted about 24 weeks, it would be interesting and important to evaluate modification in time of the parameters (not only for creatinine and urinary sKlotho), even it would have been less.

Answer: Following the recommendation of the reviewer new data showing urinary calcium over time have been included in the new version of the paper as supplementary material (Figure S1. D).

Row 343 – space between number and unit temperature.

Answer: The mistake regarding row 343 has been corrected in the revised version of the paper (line 361 in the new version of the paper with tracked changes).

Row 347 – formatting “4oC”

Answer: Done (line 365 in the new version of the paper with tracked changes).

The glycaemia was kept between 300-500 mg/dL range, in the Table 1 show a mean value of 714 mg/dL.

Answer: The procedure followed after the first week of the study was not clearly explained. In fact, our aim was not to keep blood glucose between 300-500 mg/dL but to prevent excessive body weight loss. Those rats with blood glucose higher than 500 mg/dL (the upper limit of quantification of the glucometer) and a significant body weight loss (> 10%) received additional insulin. As the glucometer used for monitoring blood glucose had an upper limit of quantification of 500 mg/L we did not know the exact levels of glucose. The exact levels of glucose were measured at the end of the study (24 weeks) by using an autoanalyzer that does not have the limit of 500 mg/dL.

Moreover, at the end of the study (week 24), glucose levels were measured in serum, not in whole blood. Levels of glucose in serum/plasma are higher than in whole blood (Tonyushkina, K., & Nichols, J. H. (2009). Journal of diabetes science and technology, 3(4), 971–980). Also, glucose levels could be increased by the isoflurane used for anaesthesia (Zuurbier CJ, et al. Anesth Analg. 2008 Jan;106(1):135-42; Yu Q et al. PLoS One. 2020 Apr 2;15(4):e0231090).

This aspect has been clarified in the material and methods section and removed from the discussion section in the new version of the paper (rows 350-357 and 212 in the new version of the paper with tracked changes)

In the material and method section id not specified how the blood sample collection were made at weeks 9 and 12.

Answer: Blood samples were only collected at week 24.

Rows 280 and 283: the sentences are not finished.

Answer: We apologize for the mistake, there was an extra “comma” at the end of both sentences.

They have been removed in the new version of the paper (rows 288 and 298 in the new version of the paper with tracked changes).

Discussion:

-          according to other data (DOI: 10.1007/s11010-018-3400-2), the plasma level of sRAGE is elevated in diabetes, contrary to the results presented in this paper, possible explanations are needed.

Answer: The relationship between sRAGE and diabetes is controversial. Previous studies found increased levels of serum sRAGE in diabetic patients, in contrast, other studies found a decreased levels of circulating sRAGE in patients with T1 and T2DM. In some cases, this discrepancy could be explained as a consequence of differences in renal function.

A new paragraph has been added to the discussion section including the reference cited by the reviewer (rows 288-297 in the revised version of the paper with tracked changes).

-          the role and urinary level of Klotho is very complex, more detailed information are needed (doi.org/10.3389/fphar.2022.931746).

Answer: We fully agree with the reviewer that the role of Klotho in CKD is very complex and its use as an urinary marker of renal function has been poorly studied.

A sentence has been added to the discussion section (rows 269-271 in the revised version of the paper with tracked changes) and the reference provided by the reviewer has been added in introduction and discussion sections (rows 78 and 271 in the revised version of the paper with tracked changes).

Reviewer 2 Report

This manuscript Role of Klotho and AGE/RAGE-Wnt/β-catenin signaling pathway on the development of cardiac and renal fibrosis in diabetes. The manuscript was well written, and the data were solid. Some minor concerns still need to be addressed before it is suitable for publication.

Minor concerns:

1. What’s the specific method were used in the statistical analysis? The author need list this information in the legend of each figure legend.

2. In fig2b, why the intensity for the last lane of GAPDH was strong?

3. There are some spelling mistakes need to be address, eg, the unit of Celsius degree.

Author Response

Reviewer 2

This manuscript Role of Klotho and AGE/RAGE-Wnt/β-catenin signalling pathway on the development of cardiac and renal fibrosis in diabetes. The manuscript was well written, and the data were solid. Some minor concerns still need to be addressed before it is suitable for publication.

Minor concerns:

  1. What’s the specific method were used in the statistical analysis? The author need list this information in the legend of each figure legend.

Answer: The specific method used in the statistical analysis has been listed in each figure legend.

  1. In fig2b, why the intensity for the last lane of GAPDH was strong?

Answer: As the reviewer knows, the differences in the housekeeping protein signal indicate different sample loads due to known errors such as pipetting or protein quantification among others. However, small differences in the intensity of GAPDH should not affect the ratio target protein/housekeeping protein.

The image has been replaced by a new one.

  1. There are some spelling mistakes need to be address, eg, the unit of Celsius degree.

The misspellings detected, such as the unit of Celsius degree, have been corrected (rows 343, 365 in the revised version of the paper with tracked changes).
